# Marine-Derived *Penicillium purpurogenum* Reduces Tumor Size and Ameliorates Inflammation in an Erlich Mice Model

**DOI:** 10.3390/md18110541

**Published:** 2020-10-29

**Authors:** Amanda Mara Teles, Leticia Prince Pereira Pontes, Sulayne Janayna Araújo Guimarães, Ana Luiza Butarelli, Gabriel Xavier Silva, Flavia Raquel Fernandes do Nascimento, Geusa Felipa de Barros Bezerra, Carla Junqueira Moragas-Tellis, Rui Miguel Gil da Costa, Marcos Antonio Custódio Neto da Silva, Fernando Almeida-Souza, Kátia da Silva Calabrese, Ana Paula Silva Azevedo-Santos, Maria do Desterro Soares Brandão Nascimento

**Affiliations:** 1Laboratory for Culture Cell, Postgraduate Program in Biotechnology (RENORBIO), Federal University of Maranhão, São Luís 65080-085, Maranhão, Brazil; damarateles@hotmail.com (A.M.T.); xaviersilva.g@gmail.com (G.X.S.); 2Laboratory of Applied Cancer Immunology, Biological and Health Sciences Center, Federal University of Maranhão, Avenida dos Portugueses, 1966 Bacanga, São Luis 65080-085, Maranhão, Brazil; leticiaprince22@hotmail.com (L.P.P.P.); sulaynebio@hotmail.com (S.J.A.G.); analuizabutarelli@gmail.com (A.L.B.); apsazevedo@yahoo.com.br (A.P.S.A.-S.); 3Immunophisiology Laboratory, Biological and Health Sciences Center, Federal University of Maranhão, Avenida dos Portugueses, 1966 Bacanga, São Luís 65080-085, Maranhão, Brazil; nascimentofrf@yahoo.com.br; 4Postgraduate Program in Adult Health (PPGSAD), Federal University of Maranhão, Avenida dos Portugueses, 1966 Bacanga, São Luís 65080-085, Maranhão, Brazil; geusabezerra@gmail.com (G.F.d.B.B.); rmcosta@fe.up.pt (R.M.G.d.C.); 5Laboratory of Natural Products for Public Health, Institute of Pharmaceutical Techonology, Oswaldo Cruz Foundation, Rio de Janeiro 21041-000, Brazil; carlatellis@far.fiocruz.br; 6Centre for the Research and Technology of Agro-Environmental and Biological Sciences (CITAB), University of Trás-os-Montes and Alto Douro (UTAD), 5001-801 Vila Real, Portugal; 7Post-graduation in Internal Medicine, State University of Campinas, Campinas 13083-887, São Paulo, Brazil; marcos_antonio456@hotmail.com; 8Laboratory of Anamotopathology, Postgraduate Program in Animal Science, State University of Maranhão, São Luis 65055-310, Maranhão, Brazil; fernandoalsouza@gmail.com; 9Laboratory of Immunomodulation and Protozoology, Oswaldo Cruz Institute, Oswaldo Cruz Foundation, Rio de Janeiro 21040-900, Brazil

**Keywords:** Ehlich’s tumor, *P. purpurogenum*, antitumor, meroterpenoids, inflammation

## Abstract

Background: This study addresses the antitumoral properties of *Penicillium purpurogenum* isolated from a polluted lagoon in Northeastern Brazil. Methods: Ethyl Acetate Extracellular Extract (EAE) was used. The metabolites were studied using direct infusion mass spectrometry. The solid Ehrlich tumor model was used for antitumor activity. Female Swiss mice were divided into groups (*n* = 10/group) as follows: The negative control (CTL−), treated with a phosphate buffered solution; the positive control (CTL+), treated with cyclophosphamide (25 mg/kg); extract treatments at doses of 4, 20, and 100 mg/kg; animals without tumors or treatments (Sham); and animals without tumors treated with an intermediate dose (EAE20). All treatments were performed intraperitoneally, daily, for 15 days. Subsequently, the animals were euthanized, and the tumor, lymphoid organs, and serum were used for immunological, histological, and biochemical parameter evaluations. Results: The extract was rich in meroterpenoids. All doses significantly reduced tumor size, and the 20 and 100 mg/kg doses reduced tumor-associated inflammation and tumor necrosis. The extract also reduced the cellular infiltration of lymphoid organs and circulating TNF-α levels. The extract did not induce weight loss or renal and hepatic toxic changes. Conclusions: These results indicate that *P. purpurogenum* exhibits immunomodulatory and antitumor properties in vivo. Thus, fungal fermentation is a valid biotechnological approach to the production of antitumor agents.

## 1. Introduction

Fungi are versatile organisms with promising therapeutic and biotechnological potential that can be found in several habitats and occupy inhospitable ecological niches in all ecosystems on the planet [1]. Several studies show that marine microorganisms are sources of unique natural products, including molecules with potential anticancer uses [2,3,4,5].

*Penicillium* fungi synthesize large amounts of known bioactive secondary metabolites [6,7], including anticarcinogenic drugs and immunosuppressive agents [8,9]. The *P. purpurogenum* species has the ability to synthesize a variety of substances with biotechnological and bioactive potential, but the strains that demonstrate this activity are mutant strains resistant to antibiotics [10,11].

*Penicillium purpurogenum* MA52 is a strain previously isolated from a polluted marine environment, the Jansen lagoon in the Northeastern Brazilian state of Maranhão [12]. This is a highly polluted environment that receives domestic effluents from the surrounding city of São Luís [12]. However, the specific characteristics of the MA52 strain that allow it to survive in this polluted environment have not yet been described. New drugs are being developed from secondary metabolites to discover less toxic and more effective compounds compared to traditional cancer therapies [13]. Some natural products have also become important nutraceuticals with cancer chemopreventive properties [14,15].

Many in vivo models are used for studying breast cancer, the most common cancer in women worldwide [16]. Breast cancer models include genetically modified animals, xenografted tumors in immune-compromised mice, chemically induced models [17], and syngeneic models like the solid Ehrlich’s tumor, which is a spontaneous and highly aggressive murine mammary adenocarcinoma [18]. This syngeneic model is immunocompetent and avoids the use of harmful chemicals, making it particularly useful for tumor and chemotherapy studies [19,20]. Considering the search for bioactive compounds capable of serving as prototypes of new antitumor drugs, the present study aims to describe the in vivo antitumor activity of *P. purpurogenum* extract compounds obtained from the MA52 strain against solid Ehrlich’s tumor.

## 2. Results

### 2.1. P. purpurogenum Extracellular Extract Contains Meroterpenoids

HPLC-DAD-UV analyses were performed to determine the *P. purpurogenum* extract profile. The obtained chromatogram at 300 nm showed 12 peaks, and all of them presented UV λ_max_ in the range of 258–282 nm, confirming the presence of meroterpenoid compounds (Figure 1 and Table 1).

The mass spectral data obtained by direct infusion in positive mode (ESI-MS/MS), presented in Figure 2, were used to confirm the chemical profile of the *P. purpurogenum* extracellular extract (EAE) and suggested the presence of five meroterpenoidal compounds (**1**–**5**) commonly produced by fungi [21] by comparing the mass spectral data with those described in the literature.

Compound **1** (purpurogemutantin) presented an experimental pseudo-molecular ion [M + H]^+^ at *m/z* 419.1674 compatible with the molecular formula C_24_H_35_O_6_. Compounds **2** (purpurogemutantin) and/or **3** (macrophorin A) showed the same pseudo-molecular ion [M + H]^+^ at *m/z* 361.1037, corresponding to the molecular formula C_22_H_33_O_4._ These compounds were previously described for *P. purpurogenum* extracts by Fang et al. [22]. Compound **4** (berkeleyacetal C, molecular formula C_24_H_26_O_8_), another meroterpenoid previously described in mutant *P. purpurogenum* by Li et al. [23], presented a pseudo-molecular ion [M + H]^+^ at *m/z* 443.1696, while compound **5** (rubratoxin B) showed a pseudo-molecular ion [M + H]^+^ at *m/z* 519.1758 corresponding to the molecular formula C_22_H_33_O_4_. Rubratoxin B is another known meroterpenoid with anticancer activity isolated by *P. purpurogenum* [24].

### 2.2. P. purpurogenum Extract Showed Activity Against Ehrlich’s Solid Tumor

In tumor growth kinetics, the results showed that, throughout the experiment, the negative control group presented an upward curve, while those treated with cyclophosphamide and the extract showed linear kinetics. The extract significantly reduced tumor growth from the eighth day compared to CLT−, while cyclophosphamide reduced tumor growth from the tenth day (Figure 3a). The kinetic values of tumor growth agree with the graph of the area under curve (AUC). The extract at doses of 4 mg/kg (422 ± 35.8 mm^2^), 20 mg/kg (389.7 ± 49.13 mm^2^), and 100 mg/kg (365.5 ± 40.47 mm^2^) presented an area smaller than that of the negative control (1240 ± 43.25 mm^2^) along with chemotherapy (599.8 ± 55.51 mm^2^) (Figure 3b). At the end of the experimental period, the extract at 4 mg/kg (0.19 ± 0.02 g), 20 mg/kg (0.19 ± 0.05 g), and 100 mg/kg (0.16 ± 0.02 g) showed a low tumor weight when compared to the negative control (0.38 ± 0.07 g) and similar weight to the positive control (0.15 ± 0.05 g) (Figure 3c).

### 2.3. Histological Results

Histopathological analysis showed the presence of tumor masses in the foot pads of animals from the groups with Ehrlich’s solid tumor. The tumor masses exhibited high cellularity and different growth patterns, with central, moderate, or very high pleomorphisms, as well as bizarre nuclear forms, the presence of one or more nucleoli, and heterogeneous chromatin patterns. The cytoplasm was eosinophilic and abundant with poorly defined limits. Occasional multinucleated giant cells (associated with pattern 3 pleomorphism) and up to two mitosis figures per 400× field were identified. Groups with tumor induction presented peritumoral, multifocal, or coalescent lymphohistiocytic inflammatory infiltrates with variable intensity. Areas of multifocal liquefaction necrosis with the accumulation of eosinophilic and amorphous cellular debris associated with neutrophilic infiltrate were also observed. Interestingly, the extract showed an inflammatory process induced by the minor tumors in the groups treated with 20 and 100 mg/kg when compared to the negative control. The extract was also demonstrated to have smaller areas of tumor necrosis, mitotic activity, and invasion compared to the negative control (Figure 4 and Table 2).

### 2.4. P. purpurogenum Extract Induced Immunomodulatory Effects

Treatment with EAE at doses of 20 and 100 mg/kg presented low number of cells in the popliteal lymph nodes (9.6 × 10^4^ ± 1.0 cells/mL and 4.4 × 10^4^ ± 0.34 cells/mL, respectively) compared to the negative control (326 × 10^4^ ± 40.83 cells/mL). Similar results were observed in the cyclophosphamide-treated animals (8.0 × 10^4^ ± 1.88 cells/mL) (Figure 5a). In the spleen cells, the negative control group (72.7 × 10^7^ ± 7.06 cells/mL) showed high cellularity compared to the animals of the Sham group (1.6 × 10^7^ ± 0.12 cells/mL). However, this difference was not observed in animals inoculated with the tumor and treated with cyclophosphamide (0.7 × 10^7^ ± 0.02 cells/mL) or extract doses of 20 mg/kg (1.6 × 10^7^ ± 0.14 cells/mL) and 100 mg/kg (1.2 × 10^7^ ± 0.04cells/mL) (Figure 5b). The marrow bone cellularity showed that the cyclophosphamide (12.3 × 10^5^ ± 3.13 cells/mL) and the extract treatment (61.0 × 10^5^ ± 8.35 cells/mL, 17.6 × 10^5^ ± 1.20 cells/mL and 4.0 × 10^5^ ± 1.08 cells/mL, respectively) prevented a high level of cellularity associated with Ehrlich’s tumor, since the groups without a tumor (sham: 4.6 × 10^5^ ± 0.77 cells/mL and EAE20: 15.44 × 10^5^ ± 5.04 cells/mL) had fewer cells than the negative control group (189.0 × 10^5^ ± 23.12 cells/mL) (Figure 5c). The results show that the extract’s effect was dose-dependent.

Cytokine quantification demonstrated that treatment was able to alter only the TNF levels (Appendix A). The animals with untreated tumors (CTL−) showed an even higher level of TNF-α concentrations (508.8 ± 66.22 pg/mL) compared to the Sham group (156.0 ± 25.14 pg/mL). The cyclophosphamide treatment significantly showed lower TNF-α levels (177.2 ± 27.67 pg/mL) compared to the negative control group. Similarly, the extract at 4, 20, and 100 mg/kg doses also showed low TNF-α concentrations versus the group of CLT− (161.7 ± 30.22, 212.4 ± 12.03 and 261.4 ± 43.16 pg/mL, respectively) in animals with solid tumors. However, the extract treatment in animals without a tumor showed a high level of seric TNF-α concentration (306.9 ± 17.93 pg/mL) compared to normal animals (Figure 5d).

### 2.5. Toxicity Studies of P. purpurogenum EAE

Daily treatment with *P. purpurogenum* ethyl acetate extract in the solid Ehrlich tumor maintained the body weights of the tumor-inoculated animals (EAE4: 2.38 ± 0.63 g; EAE20: 2.8 ± 0.76 g and EAE100: 0.96 ± 0.67 g) compared with CTL+ (−1.16 ± 0.46 g) and CTL− (−0.24 ± 0.73 g) (Figure 6a). The data show that only the group receiving cyclophosphamide had a body weight reduction compared to the Sham group (1.58 ± 0.96 g). There were no significant differences in the serum AST and ALT levels in any of the groups analyzed, although the CTL− animals showed increased average values compared to the groups treated with the extract in the presence of the tumor (Table 3). Hepatic and renal histological differences were not found between the groups. The survival rate curve showed that the saline (79.12%) and cyclophosphamide (85.72%) groups were statistically different, as expected. Importantly, the tumor-animal groups treated with the extract showed a 100% survival rate under all doses, which is significantly different compared to the controls (Figure 6b).

### 2.6. P. purpurogenum Extract Has a Cytotoxic Effect against MCF7 Cells In Vitro

*P. purpurogenum* isolated from the marine environment exhibited antitumor activity in vitro against MCF7 cells. EAE showed a concentration and time-dependent effect, as observed in the dose–response curve of the viable MCF7 cell percentage in relation to the untreated cells after 24 and 48 h of treatment (Figure 7A). The inhibitory concentration of 50% of cells (IC_50_) after treatment with EAE was 53.56 ± 1.031 and 27.22 ± 1.029 µg/mL for 24 and 48 h, respectively. The reference drug doxorubicin also presented a concentration and time-dependent effect (Figure 7B) and IC_50_ values of 37.10 ± 1.161 and 19.02 ± 1.223 µg/mL for 24 and 48 h, respectively.

## 3. Discussion

*P. purpurogenum* isolated from a marine environment is capable of secreting secondary bioactive metabolites with anticancer properties. In the present study, the extracellular extract of this fungus revealed the presence of macrophorin A, purpurogemutanthidine, and purpurogemutanthin, corroborating the previous findings by Tang et al. [25], Fang et al. [22], and He et al. [26]. These compounds showed interesting anticancer activities in vitro against cervical cancer, gastric adenocarcinoma, and breast cancer cells, while berkeleyacetal C [27,28,29] and rubratoxin B [30,31,32] showed anti-inflammatory and anticancer activity in vitro.

Meroterpenoids, which are formed by the mixed terpenoid–polyketide biosynthetic pathway, are an important class of compounds in the context of the development of new anticancer agents due to their vast structural diversity and broad spectrum of bioactivities [33]. They are known to have activities against cancer through various mechanisms, such as the blocking of cell survival pathways, activity against oxidative stress, and the induction of apoptosis [34]. When checking for the presence of compounds with anticancer activity in vitro, we questioned whether the extract isolated from the marine environment (and not described in the literature to have in vivo activity against Erhlich’s tumor) could be active.

The present study is the first to evaluate the in vivo anticancer effects of *P. purpurogenum* by employing mice inoculated with Ehrlich’s tumor cells. The results showed that the extract compounds were able to efficiently reduce the development and weight of Ehrlich’s solid tumor.

Ehrlich’s tumor exhibits strong inflammatory phenomena, including edema and inflammatory infiltrates, which are believed to play an essential role in tumor growth in various types of cancer [35,36,37,38,39]. However, anti-inflammatory effects were previously reported for meroterpenoids [40] and were also observed in this research, showing that the extract reduces inflammation, thereby reducing the tumor via an indirect route.

We found in our research that the cell line MCF7 is sensitive to the extract. This specificity for MCF-7 cells is possibly due to the cytotoxic effects caused by substances such as meroterpenoids present in the extract [22,27]. However, the research carried out by Chai et al. (2012) [10] to study the strain of the fungus *P. purpurogenum* G59 (originally derived from marine origin) against the K562 strain did not verify a cytotoxic effect or any inhibitory effect at a concentration of 1000 μg/mL. We suggest, based on the research carried out by Darsih et al. (2015) [41] changing the cytotoxic activity in another species of the same genus with the cysteine-targeted Michael acceptor as a possible pharmacophore target for fragment-based drug discovery, bioconjugation, and click reactions [41].

The extract inhibited tumor growth at the same level as the untreated group since the behavior of the animals and the survival of the group treated with the extract were similar to the results in the sham group. Thus, the extract likely acts in tumor cell proliferation, migration, and invasion.

Two factors could inhibit tumor growth: the direct effect on the tumor microenvironment caused by macrophorin A, purpurogutididine, and purpurogemutanthin identified in the extracts to result in smaller tumor size and a reduction in inflammation, thereby decreasing the tumor’s size, which may be associated with the presence of Berkeleyacetal C in the extract.

Other meroterpenoids showed the inhibition of tumor activities. Wang et al. [30] studied Guajadial, a natural dialdehyde meroterpenoid able to suppress tumor growth in human xenograft mouse models, with probable proliferation inhibition by blocking the Ras/MAPK pathway.

Wan et al. [42] demonstrated that meroterpenoids have anti-inflammatory and antioxidant activities that can reduce tumor growth. Li et al. [24] found that Berkeleyacetal C significantly inhibits the expression of inducible nitric oxide synthase (iNOS) and the production of nitric oxide by macrophages.

Berkeleyacetal C inhibits the expression and secretion of major pro-inflammatory factors and chemokines, including tumor necrosis factor-α (TNF-α) interleukin-6 (IL-6), interleukin-1β (IL-1β), macrophage inflammatory protein -1α (PImax) -1α), and the monocyte chemotactic protein-1 (MCP-1) [22,28].

Immunosuppression cells can have a pro-tumoral effect. Indeed, higher infiltration by Tregs was observed in tumor tissues, and their depletion augments antitumor immune responses in animal models [37].

There were also systemic effects observed with dose-dependent reduction in the cells of the popliteal lymph node, spleen, and bone marrow compared to the negative control group. This is in line with the low levels of TNF-α, which play an important role in the beginning and apply the activation of adhesion molecules and the expression of inflammatory mediators during inflammatory responses [43]. These pro-inflammatory mediators can cause damage to cells and tissues and also activate macrophages in various diseases associated with inflammation [44].

Increased levels of pro-inflammatory cytokines were previously associated with the development of Ehrlich’s tumor [45]. Our data also agree with those of Calixto-Campos et al. [46], Aldubayan et al. [47], and Harun et al. [48], who studied the protective role of fungal extracts against inflammatory events induced by LPS in vitro and found that all the extracts inhibited TNF-α expression. Taken together, these results show that the extracellular extract of *P. purpurogenum* has potent anticancer activity in vivo against Ehrlich’s breast adenocarcinoma and that this effect can be mediated, at least in part, by immunomodulatory mechanisms.

Another important set of observations from the present study concerns the safety of the extract. Remarkably, the *P. purpurogenum* extracellular extract was able to improve mouse survival at all doses compared to the negative control and even the cyclophosphamide, providing 100% survival at the end of the study. The extract also preserved the body weight of the animals, preventing the wasting syndrome that is often associated with cancer and intensified by chemotherapy [49,50,51,52,53], as previously reviewed [54,55].

These data suggest that the *P. purpurogenum* extract may have potential clinical use in combination therapies to prevent the loss of body weight in cancer patients. Considering that fungal extracts often display hepatic toxicity, we evaluated the ALT and AST serum levels and hepatic histology. The results did not demonstrate acute hepatic toxicity; instead, the extract did not possess the hepatic histological lesions associated with Ehrlich’s tumor and presented low serum levels of hepatic transaminases. In line with these observations, no changes were observed in renal histology. These results support the hypothesis that the extract has a favorable toxicological profile, although these observations should be confirmed and complemented by additional studies.

Thus, the results show that the extract inhibited tumor growth, compared to the negative control, at the same level as standard therapy (cyclophosphamide), even when applied at the lowest dose.

Interestingly, the activity of the extract occurred before that of the cyclophosphamide, suggesting intense antitumor activity in vivo. In addition, these findings correlate with morphological changes of the histological level, showing that *P. purpurogenum* has low mitotic activity and invasive behavior against tumor cells. The extract’s antitumor activity is associated with marked immunomodulatory effects.

## 4. Materials and Methods

### 4.1. Fermentation and Preparation of Ethyl Acetate Extracellular Extract (EAE)

*P. purpurogenum* is a marine fungus found in the coastal region (2°29′56″S 44°17′59″W) of Maranhão, Brazil. The present strain was isolated by the Mycology Laboratory of the Basic and Applied Immunology Center of the Federal University of Maranhão and deposited into the fungi collection of the Federal University of Maranhão under access code MA52. The fungus strain was grown in Potato Dextrose Agar (BDA) at 28 °C for 7 days until complete growth. After that period, superficial circles of mycelium-containing agar were further cultivated in BDA broth for fermentation at 28 °C for 21 days in a rotary shaker (Quimis, São Paulo, Brazil ) at 150 rpm. Afterward, 300 mL of the fermented broth was macerated for 48 h with 600 mL of ethanol to obtain the extracellular extract. Then, the aqueous ethanol solution was filtered and concentrated under reduced pressure to remove the ethanol, and the remaining water was extracted twice with 1:4 (*v/v*) ethyl acetate, resulting in an organic phase that was concentrated and lyophilized to obtain the ethyl acetate extract used for in vivo testing.

### 4.2. High-Performance Liquid Chromatograph Coupled to Diode-Array UV-Vis Detector (HPLC-DAD-UV)

Chromatographic analyses were performed on a HPLC-DAD-UV using a Shimadzu Nexera XR^®^ liquid chromatograph (Shimadzu, Kyoto, Japan) coupled to a UV detector with an SPDM20A diode array, a CBM20A controller, a DGU20A degasser, an LC20AD binary pump, a CTO20A oven, and an SILA20A auto-injector. Shimadzu LabSolutions Software Version 5.3 (Shimadzu, Kyoto, Japan) was used to analyze the chromatograms. DAD analysis was applied to select the optimized wavelength of the meroterpenoids in this study (300 nm). Combinations of ultrapure water (A) and methanol (HPLC grade, Tedia, Rio de Janeiro, Brazil) (B) were used as the mobile phase (initially 0% B, increasing to 20% in 8.5 min, subsequently rising to 100% of B in 68.5 min, and finally staying at this concentration up to 90 min). The HPLC column was silica-based C18 (250 mm × 4.6 mm i.d. × 5 μm particle size, Shimpack CLC-ODS, Thermo, Waltham, MA, USA). The oven was set to 50 °C, and the injection volume was 10 μL for all analyses.

### 4.3. Tandem Mass Spectrometry with Electrospray Ionization (ESI-MS/MS)

Ethyl acetate extracellular extract (EAE) was analyzed by direct infusion (ESI-MS/MS) in a Bruker Ion trap amazon SL mass spectrometer (Bruker, Billerica, MA, USA, positive mode (ESI+)). EAE (3 mg) was dissolved in certified HPLC grade methanol containing 0.1% formic acid (*v/v*) using an ultrasonic bath for 20 min. The operating conditions were 1 μL/min infusion, 4.0 kV capillary voltage, 100 °C temperature source, and cone voltage of 20–40 V. The mass spectra were recorded and interpreted by Bruker Compass Data Analysis (Version 4.2, Bruker, Billerica, MA, USA).

### 4.4. Animals

After approval by the Ethics Committee (CEUA 23115.11239/2017-70), female Swiss mice (*n* = 70), weighing 25–30 g and aged between 3 and 4 months were provided by the Federal University of Maranhão (UFMA). Animals were kept in a room with a controlled temperature of 22 ± 3 °C, with 50 ± 15% relative humidity, a 12 h light/dark photoperiod, and food and water ad libitum. The animal experiments were conducted according to the animal welfare guidelines of the National Council for the Control of Animal Experimentation (CONCEA). Every effort was made to reduce the number of animals used and their discomfort.

### 4.5. Ehrlich Solid Tumor Model

Animals were anesthetized with ketamine/xylazine (120–150 mg/kg). The Ehrlich ascitic carcinoma was maintained in the mice via intraperitoneal injections of 2 × 10^6^ cells [56]. To induce the solid tumor, transplantable neoplastic cells with 7 days of ascitic evolution were aspirated, and 200 µL of the cell suspension at 2 × 10^6^ cells/mL was injected into the left posterior foot pad. The cells were found to be more than 99% viable by a Trypan blue exclusion method. The experimental treatment began 24 h after the tumor implantation [57].

### 4.6. Treatment Groups

The animals were separated into two groups: those with tumor inoculation for antitumor activity and those without tumor inoculation used to evaluate the extract toxicity. The tumor inoculation group was subdivided into six subgroups (*n* = 10): the negative control subgroup (CTL−) with tumor induction treated with phosphate buffered solution (PBS); the positive control subgroup (CTL+) treated with cyclophosphamide at a dose of 25 mg/kg; and the subgroups treated with extracts at doses of 4 mg/kg (EAE4 + Tumor), 20 mg/kg (EAE20 + Tumor), and 100 mg/kg (EAE100 + Tumor). To determine the acute toxicity, the two groups without tumor induction were treated with PBS (Sham) or with the extract at a medium dose of 20 mg/kg (EAE20). All treatments were performed intraperitoneally 24 h after tumor inoculation over fifteen days, and the volume administrated was 100 µL. After the treatment, the animals were randomly selected. A portion was euthanized (*n* = 5) for biological analysis, and another portion (*n* = 5) was used for the quality-of-life/survival test. Euthanasia was performed via the intraperitoneal administration of 120 mg/kg ketamine and 150 mg/kg xylazine (2:1 solution) [58] before complete necropsy.

### 4.7. Tumor Development Assay

The Ehrlich solid tumor model was applied to evaluate antitumoral activity. The paw volume was determined before and after the injection of Ehrlich tumor cells using a digital caliper at 48 h intervals. The volume was calculated by multiplying the thickness, width, and length measurements of the paw with tumor presented as mm^3^. After the treatment performed over fifteen days, the animals of each group were randomly chosen (*n* = 5), euthanized, and the paws with tumors were removed and weighed. The area under the curve was calculated from the kinetics graph obtained from the tumor-inoculated paw development data using the GraphPAd Prism software (Version 7.00, GraphPad Software, San Diego, CA, USA).

### 4.8. Histopathological Analyses

Immediately after euthanasia, the Ehrlich’s tumor tissue (and matched normal tissue from the Sham groups), livers, and kidneys were removed and fixed in 10% neutral buffered formalin. The sample sections were stained with hematoxylin and eosin solution and analyzed by a single researcher with expertise, blinded to the experimental groups. For the tumor samples, the following parameters were analyzed: inflammatory infiltrate distribution (focal, multifocal, diffuse, or peripheral); inflammatory infiltrate intensity (scores: absent 0, light 1, moderate 2, and intense 3); necrosis (scores: absent 0, focal 1, focally extensive or multifocal 2, and diffuse 3); mitotic figures (scores: no mitotic figures 0, occasional mitoses 1, single mitotic figure per 400× field 2, and two mitotic figures per 400× field 3); cellular pleomorphism (scores: monomorphic tumor cells 0, minimal intercellular variation 1, variations in nuclear size and shape 2, and major variations with bizarre nuclei 3); and tumor invasion (scores: well-defined borders with no obvious invasion 0, well-defined borders with minimal invasion of adjacent tissues 1, poorly-defined borders with marker invasion 2, and unrecognizable borders with multiple tumor foci 3) [59].

### 4.9. Lymphoid Organ Cellularity

To obtain and quantify cells in the popliteal lymph node and spleen, these solid organs were removed and macerated in 1 mL PBS. To obtain bone marrow cells, the femur was removed and perfused with 1 mL of PBS. Next, 90 μL of lymph node, spleen, and bone marrow cell suspensions were added to 10 μL violet crystal, and the cells were counted in a Neubauer chamber with the aid of a common light optical microscope (Zeiss, Oberkochen, Germany) [60].

### 4.10. Blood Samples

Blood samples were obtained by cardiac puncture. The samples were then centrifuged at 5000 rpm for 10 min. The serum was separated and stored in aliquots at −80 °C until needed. Prior to the assay, the samples were thawed at room temperature [59,60].

### 4.11. Cytokine Quantification

Blood serum was used for the quantification of interleukin-2 (IL-2), interleukin-4 (IL-4), interleukin-6 (IL-6), interferon-γ (IFN-γ), tumor necrosis factor (TNF-α), interleukin 17A (IL-17A), and interleukin-10 (IL-10) by flow cytometry with FACS Calibur equipment (BD Biosciences, San Jose, CA, USA) using the BD™ Cytometric Bead Array (CBA) cytokine kit Mouse Th1/Th2/Th17 (BD Biosciences, San Jose, CA, USA) following the manufacturer’s recommendations.

### 4.12. Extract Toxicity

For the assessment of acute toxicity, we considered the weight of the animals during the treatment (the animals were weighed daily for 15 days). Weight variation was verified before tumor inoculation until the last day of treatment. We also verified the hepatic parameters of the fungus extract. Blood serum was used to perform the biochemical measurement of glutamic-oxalic transaminase (TGO) and glutamic-pyruvic transaminase (TGP) through colorimetric analyses using Labtest kits, following the manufacturer’s guidelines. The data were obtained on a visible spectrum plate reader (Lab. Syftemf Multi Skan EX, Version 1.00, Waltham, Ma, USA). Histopathological analysis of the liver and kidney was performed. The parameters analyzed in the liver were the presence of mitotic figures, caryatia (more than 10% hepatocytes with nuclei twice the size of normal hepatocytes), and the presence or absence of necrosis. Hepatitis was classified as mild (hyperplasia of Küpfer cells and/or occasional microabscesses or mild focal periportal leukocytic infiltration) or moderate (multifocal to diffuse leukocytic infiltration in multiple portal spaces or centrilobular veins). Hepatocellular vacuolar degeneration was also classified as mild (restricted to the periportal and/or centrilobular areas) or moderate (extending to the midzonal areas). In the kidney, the presence of tubular degeneration, defined as the swelling of the cells of the outlined proximal or distal tubules, the necrosis of isolated tubular cells, or the loss of cell vesicles in the tubular lumen (bleeding), was evaluated [61]. The quality of life/survival of the animals was verified for 30 days after treatment, following a quality-of-survival protocol in which some of the animals that suffered damage were assessed using a pain scale. Using this scale, if the animal showed three or more potential signs associated with pain or discomfort, they were kept under surveillance for more than 72 h and then euthanized by excess anesthetic [62].

### 4.13. In Vitro Cytoxicity against MCF7 Cells

The cell line MCF-7 (ATCC^®^ HTB-22™) was kindly donated by the Laboratório de Tecnologia de Anticorpos Monoclonais, Biomanguinhos, Fiocruz-RJ. The cells were grown in a sterile bottle containing modified Dulbecco’s Modified Eagle’s Medium (DMEM) (Invitrogen, Carlsbad, CA, USA), supplemented with 10% fetal bovine serum with 2 mM glutamine, 100 U/mL penicillin, and 100 µg/mL streptomycin. Then, the cells were incubated at 37 °C in a humid atmosphere containing 5% CO_2_. For the experiments, the monolayer cells were trypsinized with a 0.25% (*w/v*) trypsin solution–0.03% (*w/v*) EDTA. The lyophilized extract was dissolved in DMSO 0.1%, and the solution was filtered through a 0.2 μm pore syringe filter and stored at −20 °C until use. The cultured cells were treated with concentrations between 1000 and 7.8 μg/mL, obtained by serial dilution of the extracts (1:2) for 24 and 48 h. In the viability assay, the cells (5 × 10^4^ cells/mL) were grown in 96-well plates in the presence or absence of the extract for 24 and 48 h. In total, 10 μL of 3-(4,5-dimethylthiazol-2-yl)-2,5-diphenyltetrazolium bromide (MTT) at 5 mg/mL was added to each well. The cells were incubated in a CO_2_ chamber for 3 h with protection from light. Then, the medium was removed, and 100 μL of DMSO was added to dissolve the formazan crystals. The absorbance at 570 nm was measured with a Biochrom EZ Read 400 spectrophotometer (Biochrom, Cambridge, UK). The data were then normalized, and the inhibitory concentration for 50% of the cells was calculated from the non-linear regression of the percentage of viable cells versus the log of the concentration of the treatment using the GraphPAd Prism software (Version 7.00, GraphPad Software, San Diego, CA, USA) [63].

### 4.14. Statistical Analysis

The results were expressed as the mean ± standard error of means (SEM or S.D.). The differences were submitted to an analysis of variance (one-way or two-way ANOVA) followed by a Newman–Keuls test and a Student’s t-test using the GraphPad Prism software, version 7.0. To evaluate the survival curve, a Kaplan–Meier curve was used, and the statistical analysis was performed by a Log-Rank test. The significance level for rejection of the null hypothesis was 5% (*p* < 0.05).

## 5. Conclusions

Overall, the present results indicate that the *P. purpurogenum* extract has potent in vivo anticancer activity against Ehrlich’s solid tumor, suggesting the presence of immunomodulatory mechanisms. The chemical components present in the extract may serve as lead compounds for the development of new compounds with antitumor effect and immune response modulators. The extract was well-tolerated and improved animal survival compared to cyclophosphamide, suggesting a favorable toxicity profile and potential applications in combination therapies. Further preclinical studies are required to clarify the potential uses of *P. purpurogenum* and better understand the production of bioactive metabolites in fungi for biotechnological applications.

## Figures and Tables

**Figure 1 marinedrugs-18-00541-f001:**
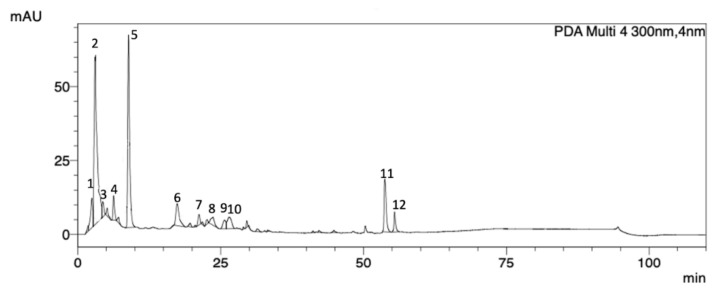
HPLC-DAD-UV chromatogram (300 nm) obtained from the *Penicillium purpurogenum* extract showing 12 peaks of which the UV spectral data (λ max) are characteristic of meroterpenoid compounds.

**Figure 2 marinedrugs-18-00541-f002:**
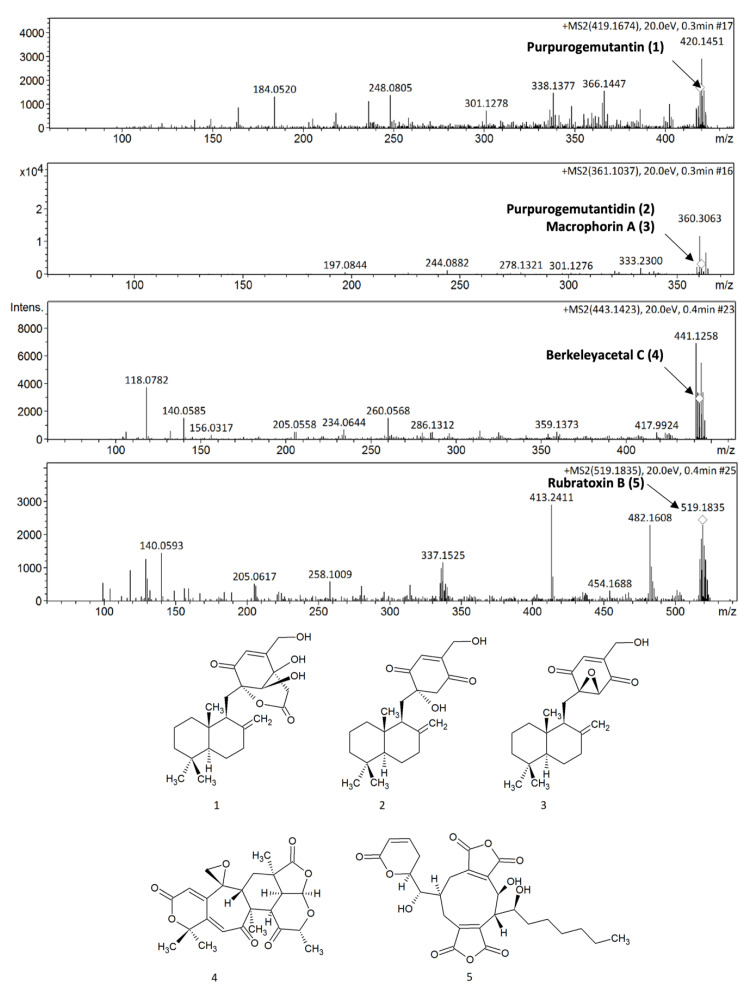
Mass spectra obtained for the tentative identification of meroterpenoids **1**–**5**.

**Figure 3 marinedrugs-18-00541-f003:**
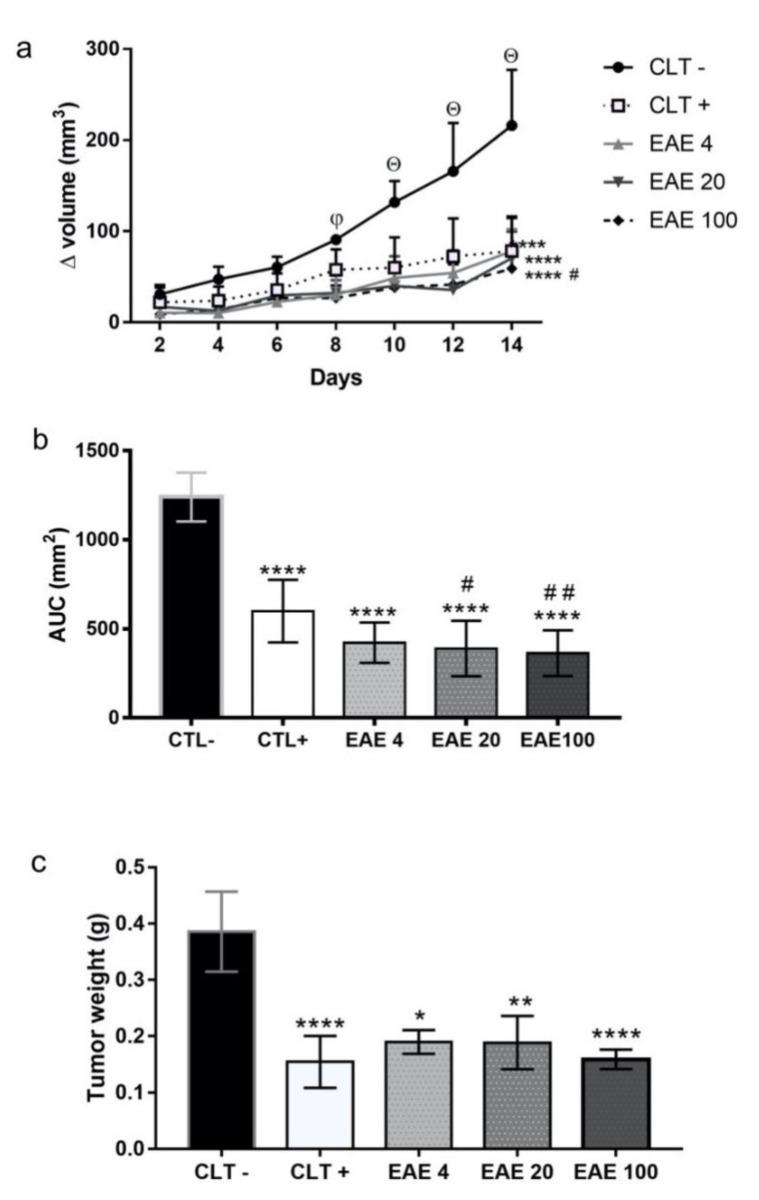
Effect of the *Penicillium purpurogenum* extract on the development of Ehrlich’s solid tumor. (**a**) The kinetic of the paw volume inoculated with Ehrlich’s tumor followed by intraperitoneal treatment with phosphate buffer solution (CTL−), cyclophosphamide 25 mg/kg (CTL+), and extracts with a concentration of 4 mg/kg (EAE4), 20 mg/kg (EAE20), and 100 mg/kg (EAE10) at 24 h intervals, with the animals being euthanized on day 15. (**b**) The area under the curve (AUC) calculated from the volume growth kinetics. (**c**) The average weight of the paws with a tumor was determined in the groups after treatment. Values are expressed as the mean ± standard error of means (SEM) deviation and analyzed by analysis of variance (one-way or two-way ANOVA) with * *p* < 0.05, ** *p* < 0.005, and **** *p* < 0.0001 relative to the negative control (CTL−); # *p* < 0.05, ## *p* < 0.01 when compared to the positive control (CTL+); and Θ *p* < 0.05 shows a difference in tumor growth for the other groups; φ shows that, on the eighth day, only the extract inhibited tumor growth.

**Figure 4 marinedrugs-18-00541-f004:**
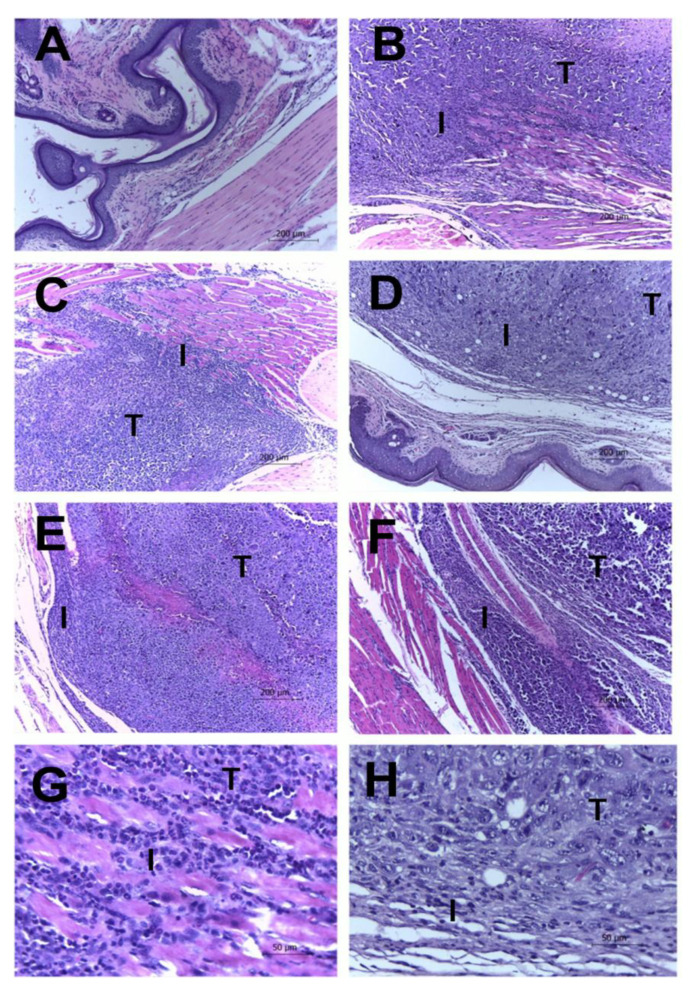
Tumor and leukocyte infiltration in Ehrlich tumors with and without treatments. Leukocyte infiltrates in Ehrlich tumors. Swiss mice were inoculated in the paw with 2 × 10^6^ Ehrlich tumor cells and treated daily with EAE extract intraperitoneally. At the end of the fifteen days of treatment, the animals were euthanized, and their feet were amputated, weighed, and fixed. Histological sections were stained with Hematoxylin–Eosin. In the photos, it is possible to see the tumor cells (indicated by the letter T) and the inflammatory infiltrate (indicated by letter I) present in the paws of the Sham (**A**), CLT− (**B**), and CLT+ (**C**) groups (100× total magnification). The animals treated with the extract showed a decrease in the infiltrate and necrosis are EAE doses of EAE 4 (**D**), EAE 20 (**E**), and EAE 100 (**F**) (100× total magnification). The Sham group is shown in panel **G,** and the animals treated using the extract with a dose of EAE 100 are shown in panel **H** (400× total magnification).

**Figure 5 marinedrugs-18-00541-f005:**
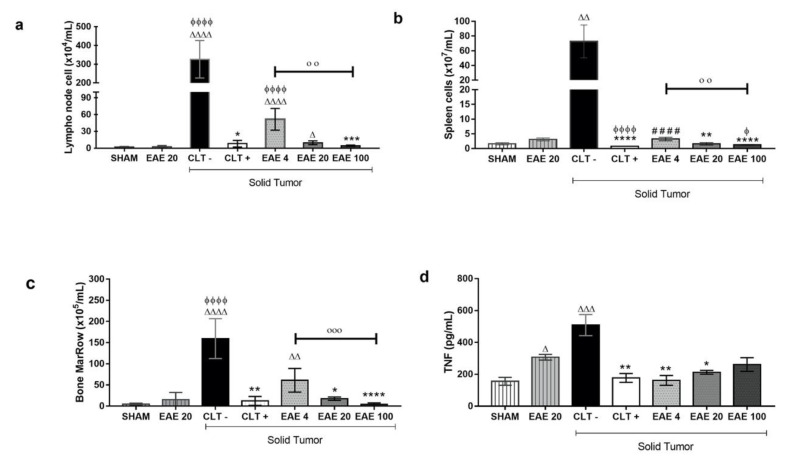
Immunomodulatory effects of *Penicillium purpurogenum* ethyl acetate extract. (**a**–**d**) The results are expressed as the mean ± standard deviation of the total lymph node cell count; bone marrow and splenocytes obtained from the group without tumors treated with saline solution (Sham) from the groups with solid Ehrlich tumors treated intraperitoneally with extracts at doses of 4, 20, and 100 mg/kg, respectively (EAE4, EAE20, and EAE100); positive and negative control-administered cyclophosphamide (CLT+) and saline solution (CTL−), respectively. Blood serum was used to quantify the tumor necrosis factor (TNF-α). The data were submitted to statistical analyses via Kruskal–Wallis and Dunn multiple comparison tests, with a significance of ∆ *p* < 0.05, ∆∆ *p* < 0.005, ∆∆∆ *p* < 0.0005, and ∆∆∆∆ *p* < 0.0001 in relation to the Sham: ϕ *p* < 0.05, ϕϕϕϕ *p* < 0.0001 when compared to EA20 without tumor; #### *p* < 0.0001 when compared to CLT+, * *p* < 0.05, ** *p* < 0.005, *** *p* < 0.0005, and **** *p* < 0.0001 when compared to CLT−; and οο *p* < 0.0001 οοο *p* < 0.0001 when comparing the extracts with each other.

**Figure 6 marinedrugs-18-00541-f006:**
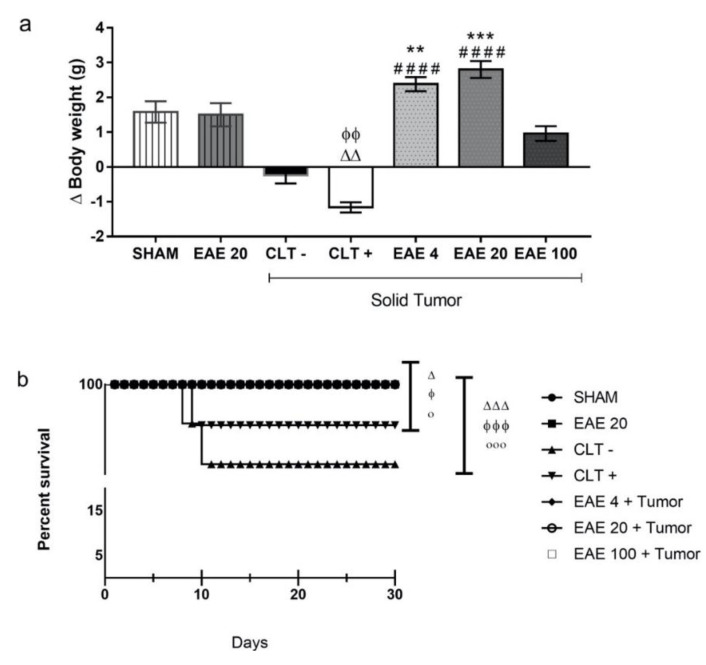
Effect of the toxicity of the extracellular extract of *Penicillium purpurogenum* ethyl acetate extract (EAE). The groups without tumor induction were treated with saline (sham), while the groups with induction received the extract at doses of 4, 20, and 100 mg/kg (EAE4, EAE20, and EAE100), as well as saline (CTL−) and cyclophosphamide (CTL+). The graphs show (**a**) the animals’ body weights in the final treatment. (**b**) After the treatment of the animals, the animals remained under observation for thirty days. The data represent the mean ± SEM. The difference was statistically analyzed by a Kruskal–Wallis and Dunn’s multiple comparison test, with significance of ∆ *p* < 0.05, ∆∆ *p* < 0.005, and ∆∆∆ *p* < 0.0005 in relation to the Sham, ϕ *p* < 0.05 ϕϕ *p* < 0.005, and ϕϕϕ *p* < 0.0005 when compared to EA20 without a tumor, ο *p* < 0.05 when compared to CLT+, and οοο *p* < 0.0005 when compared to CLT−, * *p* < 0.05, ** *p* < 0.005, and #### *p* < 0.0001.

**Figure 7 marinedrugs-18-00541-f007:**
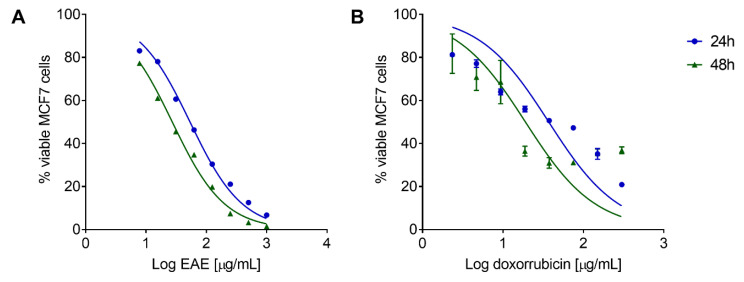
Dose–response curve of *Penicillium purpurogenum* ethyl acetate extract (EAE) for MCF7 cell viability. Data represent the means ± SD of the viable MCF7 cell percentage in relation to the untreated cells after 24 and 48 h of treatment with EAE (**A**) and doxorubicin (**B**).

**Table 1 marinedrugs-18-00541-t001:** Meroterpenoids of the crude extract of *Penicillium purpurogenum* and their retention times, relative composition, and UV data.

Peak	Rt (min)	Area %	UV (λ_max_, nm)
1	2.46	4.89	258
2	3.37	37.30	259
3	4.36	2.00	264
4	6.27	2.90	283
5	8.87	26.96	272
6	17.38	5.63	282
7	21.15	1.52	265
8	23.65	2.13	271
9	26.27	2.02	276
10	26.48	3.82	277
11	53.81	8.31	278
12	55.41	2.48	278

Rt: retention time.

**Table 2 marinedrugs-18-00541-t002:** Ehrlich tumor histopathological scores (mean ± SD) of groups treated with saline (CTL−), cyclophosphamide (CTL+), or *Penicillium purpurogenum* ethyl acetate extract at 4 mg/kg (EAE4), 20 mg/kg (EAE20), and 100 mg/kg (EAE100).

	Pleomorphism	Necrosis	Mitosis Figures	Inflammation	Invasion
CLT−	2.4 ± 0.511	2.4 ± 0.843	1.0 ± 1.155	2.8 ± 0.421	2.0 ± 0.666
CLT+	0.2 ± 0.421 *	0.2 ± 0.421 *	0.0 ± 0.000 *	1.0 ± 1.333 *	0.6 ± 0.843 *
EAE4	2.2 ± 1.122	1.2 ± 1.033	1.0 ± 1.155	1.4 ± 1.075	1.0 ± 0.667
EAE20	2.0 ± 1.155	0.6 ± 1.265 *	0.2 ± 0.421	0.8 ± 0.788 *	1.0 ± 0.667
EAE100	0.8 ± 1.033 *	0.2 ± 0.421 *	0.0 ± 0.000 *	0.6 ± 0.843 *	0.4 ± 0.843 *

Scores: 0 (absent), 1 (weak), 2 (moderate), and 3 (intense); the result was calculated by the mean of the scores; * *p* < 0.05 relative to the negative control (CLT−). *N* = 5/group.

**Table 3 marinedrugs-18-00541-t003:** Effect of *Penicillium purpurogenum* ethyl acetate extract on ALT and AST serum levels (mean ± SEM) in all experimental groups.

	ALT (U/L)	AST (U/L)
Sham	125.2 ± 103.5	125.2 ± 103.5
EAE20 *	122.5 ± 101.5	295.3 ± 236.1
CLT−	207.9 ± 181.8	438.3 ± 328.5
CLT+	438.3 ± 328.5	293 ± 131.8
EAE4 + Tumor	50.12 ± 33.38	178.6 ± 95.15
EAE20 + Tumor	68.3 ± 57.5	185.1 ± 111.2
EAE100 + Tumor	48.88 ± 21.68	179.1 ± 61.28

*N* = 5 for each group. The one-way ANOVA showed no statistical differences between the groups. * Tumor free group treated with an extract at a dose of 20 mg/kg. The groups were treated with saline (CTL−), cyclophosphamide (CTL+), and *Penicillium purpurogenum* ethyl acetate extract at 4 mg/kg (EAE4 + Tumor), 20 mg/kg (EAE20 + Tumor), and 100 mg/kg (EAE100 + Tumor).

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
