# Peer review of "Marine-Derived Penicillium purpurogenum Reduces Tumor Size and Ameliorates Inflammation in an Erlich Mice Model"

_marinedrugs, 2020, doi:10.3390/md18110541_

Round 1

Reviewer 1 Report

The paper is an excellent onco-pharmacological evaluation of an meroterpenoid rich extract from Penicillium purpurogenum utilizing many end points and test systems. The authors have found that the tested extract significantly reduced tumor size, ameliorated the tumour-associated inflammation and necrosis. Further the cellular infiltration of lymphoid organs and the circulating TNF-α levels were modulated as well. A pilot toxicological evaluation showed that the tested extract did not induce weight loss and did not alter the biochemical surrogate marker for renal or hepatic toxic injury. The manuscript is well written and could be considered for publication in the journal.

Minor remarks:

in the abstract, row 44, the authors have erroneously stated that the animals were treated with cyclophophamide at a dose of 25mg/mL, and it should be replaced by 25 mg/kg

The binominal name of the fungal species Penicillium purpurogenum should be written in italic, throughout the manuscript.

Author Response

We would like to thank you for your suggestions about this manuscript

In order to your requests, we have changed the manuscript

Minor remarks:

in the abstract, row 44, the authors have erroneously stated that the animals were treated with cyclophophamide at a dose of 25mg/mL, and it should be replaced by 25 mg/kg

Answer: We agree with the observation, the dose unit will be changed to mg/kg in the abstract.

The binominal name of the fungal species Penicillium purpurogenum should be written in italic, throughout the manuscript.

Answer: We appreciate the correction. We will correct the writing of the species name for the italic form

Reviewer 2 Report

In overall, this study is descriptive and without any molecular mechanisms.

  1. Author identified several components in the extract using mass spectrometry analysis, however they did not determine which compound provide the major contribution for the cancer treatment.
  2. Authors may need to provide the concentration and ratio of the major components.
  3. The first time named, the binomial name must be written in full followed by the authorities and the family between parentheses.
  4. Authors may need to perform in vitro experiments to determine the mechanism of cytokine production and and their cellular functions.
  5. Need to provide the results of cytotoxic analysis for the extract.

Author Response

We would like to thank you for your suggestions about this manuscript

In order to your requests, we have changed the manuscript

In overall, this study is descriptive and without any molecular mechanisms.

  1. Author identified several components in the extract using mass spectrometry analysis, however they did not determine which compound provide the major contribution for the cancer treatment.

Answer: We agree that the experiment does not study the molecular mechanisms of the extract components. However, the work presents a preclinical study of P. purpurogenum extract in active antitumor and presents the related immunophysiological mechanism in this model. Considering the importance of drugs capable of interacting with the host's immunity and having an antitumor activity, we believe in the potential of the compounds and do not exclude possible synergism between them.

2. Authors may need to provide the concentration and ratio of the major components.

Answer: We added a new chemical analysis by HPLC-DAD-UV and provided with the relative percentages of all compounds in the crude extract (Table 2).

3. The first time named, the binomial name must be written in full followed by the authorities and the family between parentheses.

Answer: It was corrected in the manuscript.

4. Authors may need to perform in vitro experiments to determine the mechanism of cytokine production and and their cellular functions.

Answer: In this work, we observed a decrease of TNF-alfa that may indicate an immunological modulation activity of P. purpurogenum. This immunomodulatory effect will be evaluated in detail in further study in an appropriated model to determine the mechanisms of cytokine modulation, since the model used in this manuscript is not the most suitable for this type of study. We thank you for the comment and agreed with you, and this study is already in our agenda.

5. Need to provide the results of cytotoxic analysis for the extract.

Answer: Cytotoxicity analysis results was added to the manuscript (Figure 7)

Reviewer 3 Report

This manuscript reports the in vivo anticancer properties of the fungus Penicillium purpurogenum. This work evaluated in vivo anticancer activity an ethyl acetate crude extract of the fungus in a solid Ehrlic tumor model, and the active compounds were tentatively identified by LC/MS. It was found that the crude extract of the fungus P. purpurogenum exhibited immunomodulatory and antitumor properties in vivo. Experiments in vivo anticancer activity in mice were well carried out, however, the weakness of this work is the identification of the active compounds. It would be nice if 1H NMR spectrum of a crude extract, which shows signals of meroterpenoids 1-5, is submitted with this manuscript.  This manuscript is useful for readers in the field, and it is therefore recommended for publication after minor revision. In order to improve this manuscript, please consider the comments and suggestions which are listed below.

  1. “P. purpurogenum species has 63 the ability to synthesize a variety of substances….”; please use italic letters for the scientific name. Please also correct at “P. purpurogenum MA52 is a strain previously….”, and throughout the manuscript.
  2. Introduction; this work reports anticancer activity of the Penicillium fungus, so it is worth mentioning that the Penicillium fungus was found to produce potent anticancer compounds, please see RSC Adv., 2015,5, 70595-70603.
  3. The weakness of the work is the identification of the active compounds. At least, the 1H NMR spectrum of a crude extract is necessary to confirm the presence of compounds 1-5.

Author Response

We would like to thank you for your suggestions for this manuscript.

In order to your requests, we are sending the updated version.

This manuscript reports the in vivo anticancer properties of the fungus Penicillium purpurogenum. This work evaluated in vivo anticancer activity an ethyl acetate crude extract of the fungus in a solid Ehrlic tumor model, and the active compounds were tentatively identified by LC/MS. It was found that the crude extract of the fungus P. purpurogenum exhibited immunomodulatory and antitumor properties in vivo. Experiments in vivo anticancer activity in mice were well carried out, however, the weakness of this work is the identification of the active compounds. It would be nice if 1H NMR spectrum of a crude extract, which shows signals of meroterpenoids 1-5, is submitted with this manuscript.  This manuscript is useful for readers in the field, and it is therefore recommended for publication after minor revision. In order to improve this manuscript, please consider the comments and suggestions which are listed below.

  1. “P. purpurogenum species has 63 the ability to synthesize a variety of substances….”; please use italic letters for the scientific name. Please also correct at “P. purpurogenum MA52 is a strain previously….”, and throughout the manuscript.

Answer: The manuscript was carefully revised.

2. Introduction; this work reports anticancer activity of the Penicillium fungus, so it is worth mentioning that the Penicillium fungus was found to produce potent anticancer compounds, please see RSC Adv., 2015,5, 70595-70603.

Answer: The paper suggested was added to the manuscript (Line 280).

3. The weakness of the work is the identification of the active compounds. At least, the 1H NMR spectrum of a crude extract is necessary to confirm the presence of compounds 1-5.

Answer: To confirm the meroterpenoid-rich profile of P. purpurogenum extract, it was included a chromatogram (300nm) of the extract showing 12 compounds with UV absorption characteristics of this class of compounds. Only 5 were tentatively identified by mass spectral data in comparison with literature (Table 2)
